



# A simple snow temperature index model exposes discrepancies between reanalysis snow water equivalent products

Aleksandra Elias Chereque[1], Paul J. Kushner[1], Lawrence Mudryk[2], Chris Derksen[2], Colleen Mortimer[2]

[1]Department of Physics, University of Toronto, 60 St. George St., Toronto ON, M5S 1A7, Canada
[2]Climate Research Division, Environment and Climate Change Canada, 4905 Dufferin St, North York, ON M3H 5T4, Canada

*Correspondence to*: Aleksandra Elias Chereque (aleksandra.eliaschereque@mail.utoronto.ca)

**Abstract** Current global reanalyses show marked discrepancies in snow mass and snow cover extent for the
Northern Hemisphere. Here, benchmark snow datasets are produced by driving a simple offline snow model, the Brown Temperature Index Model (B-TIM), with temperature and precipitation from each of three reanalyses. B-TIM offline snow performs comparably to or better than online (coupled land-atmosphere) reanalysis snow when evaluated against *in situ* snow measurements. Sources of discrepancy in snow climatologies, which are difficult to isolate when comparing online reanalysis snow products amongst themselves, are partially elucidated by
separately bias-adjusting temperature and precipitation in B-TIM. Interannual variability in snow mass and snow spatial patterns is far more self-consistent amongst offline B-TIM snow products than amongst online reanalysis snow products, and specific artifacts related to temporal inhomogeneity in snow data assimilation are revealed in the analysis. B-TIM, released here as an open-source, self-contained Python package, provides a simple benchmarking tool for future updates to more sophisticated online and offline snow datasets.

**1 Introduction**

Terrestrial snow is a highly variable component of the cryosphere that responds to and feeds back on anthropogenic global warming via snow albedo (e.g. Betts et al., 2014; Thackeray et al., 2018). At its maximum, snow covers up to 50% of the Northern Hemisphere land surface (Robinson & Frei, 2000) and it controls a wide range of hydrological, ecological, and socio-economic systems (Bokhorst et al., 2016). Snow variability and trends
have been monitored over several decades (Doesken & Judson, 1997; Mudryk et al., 2020; Robinson, 1989) with regular reporting, such as in the annual National Oceanic and Atmospheric Administration (NOAA) Arctic Report Card. Despite this attention to snow, there are marked discrepancies in historical snow estimates from available products, leading to gaps in our understanding of snow across a range of spatial scales, from point to watershed to hemispheric (Magnusson et al., 2015; Mudryk et al., 2015). Many factors lead to these discrepancies, making
it a challenge to identify a single authoritative dataset for historical snow water equivalent or related variables. Furthermore, the simplest snow models can perform comparably to the most complex snow models against the available *in situ* observations (Boone & Etchevers, 2001; Essery et al., 2013; Magnusson et al., 2015; Menard et al., 2021). For this reason, "offline" datasets generated with Temperature Index Models (TIMs), snow models forced only by air temperature and precipitation that do not represent coupling of snow to the land-atmosphere
system, are still maintained (e.g., Hock, 2003; Ohmura, 2001; Sturm, 2015, Walter et al., 2005). Recent studies have advocated for the use of multi-product ensembles spanning a range of complexity (including offline snow



models, land surface data assimilation systems, and coupled atmosphere-land reanalysis systems) and a range of snow schemes from single-layer to multilayer snow modules embedded inside comprehensive land surface models. These ensembles can then be used to characterize snow climatology and trends (e.g. Mudryk et al., in

discussion), evaluate new snow datasets, or to quantify uncertainties (Essery, 2015; Kim et al., 2021; Mudryk et al., 2015). Methods to evaluate the quality of potential ensemble members are actively being explored.

In this study, we use an offline TIM to investigate discrepancies in snow water equivalent (SWE) and snow cover extent (SCE) in online reanalysis snow products. We use an updated version of the Brown et al. (2003) TIM,

hereafter called the "B-TIM", whose broad applicability and extensive legacy at Environment and Climate Change Canada (ECCC) motivate its use. The model was initially developed to provide a first guess field for a gridded snow analysis using forcing from the European Centre for Medium-range Weather Forecasting (ECMWF) Reanalysis 15 (ERA-15). The snow analysis was used to evaluate global climate model output from AMIP II. Later, using forcing from numerical weather forecasts to run the B-TIM, the model was incorporated into the

Canadian Meteorological Centre's (CMC's) daily snow depth analysis (Brown & Brasnett, 2010). This dataset continues to be used as a validation product for other studies (Kim et al., 2021, Zhang et al., 2021). Until recently, the standalone version of B-TIM that is internally available at ECCC was coded in Fortran 77 and forced with temperature and precipitation forcing from ERA-Interim (Dee et al., 2011). The ERA-Interim version participated in several ensemble studies (e.g. Brown et al., 2010; Brown & Robinson, 2011; Mortimer et al., 2020), provided

hemispheric snow mass estimates for the NOAA Arctic Report Card (2017 edition to 2020 edition; e.g., Mudryk et al., 2020), but has been superseded by a version forced with ERA5. In addition to updated forcing, following ECCC's push to provide transparent and reproducible open-source climate assessment tools based on FAIR principles (Environment and Climate Change Canada, 2021), we are motivated to release B-TIM as an open-source code following updated coding standards.


Discrepancies in snow from "online" coupled reanalysis arise from many sources, including inconsistencies of snow data assimilation schemes, underlying snow and land-surface component model differences, atmospheric model differences, differences in processes governing the coupled surface-energy balance, and interactions between all these factors. To highlight one example which we will discuss in more detail below, while the

assimilation of snow data may improve instantaneous estimates of snow depth, there is evidence that significant time series discontinuities may result as data streams change (as in ERA5; Mortimer et al., 2020). Like any offline TIM, B-TIM does not assimilate snow data, does not capture surface energetics, and features no coupling between snow and the land-atmosphere state. B-TIM offline snow products, provided they are suitably validated, can thus isolate the role of meteorological driving from issues related to data assimilation, model bias, and errors arising

from coupling, all of which can be sources of discrepancy for more sophisticated snow datasets. Our approach offers a useful framework to identify spurious datasets generated from complex snow modeling approaches.

We document an updated B-TIM algorithm (Sect. 2), which we release here as an open source, self-contained Python repository. We then use the B-TIM to generate offline SWE and snow cover extent using temperature and

precipitation forcing from the global reanalyses ERA5, JRA-55, and MERRA-2 for 1980-2020. Through validation with *in situ* data, we compare the realism of offline B-TIM and online coupled reanalyses (Sect. 3).



The validation dataset, which is also used in Mudryk et al. (in discussion) to evaluate a suite of 23 gridded SWE products, is described in Mortimer et al (in discussion). Our main scientific work here, which is described in Sect. 3, will be to use B-TIM to characterize and explain discrepancies amongst online reanalysis snow products'
climatological characteristics and interannual variability. This analysis will include the use of bias-adjusted temperature and precipitation forcing in B-TIM to elucidate sources of discrepancy. We discuss results and conclusions in Sect. 4.

## 2 Data and Methods

### 2.1 The B-TIM snow model

The calculations described in this section, which can also be seen in the schematic in Figure 1, comprise version 1.0.0 of the B-TIM (10.5281/zenodo.10044950). Relative to Brown et al. (2003), we provide updated documentation, and changes to some constants, reflected in the code and its parameters. Physical constants and parameter values can be found in Table 1. At a given time step, we denote the initial snow depth and density by $D_i$ and $\rho_i$. $SWE_i$ is the initial time step's snow water equivalent, calculated as $SWE_i = \rho_i D_i$ with units of kg m$^{-2}$.
$^2$. All densities have standard units (kg m$^{-3}$), and snow depths have units of metres.

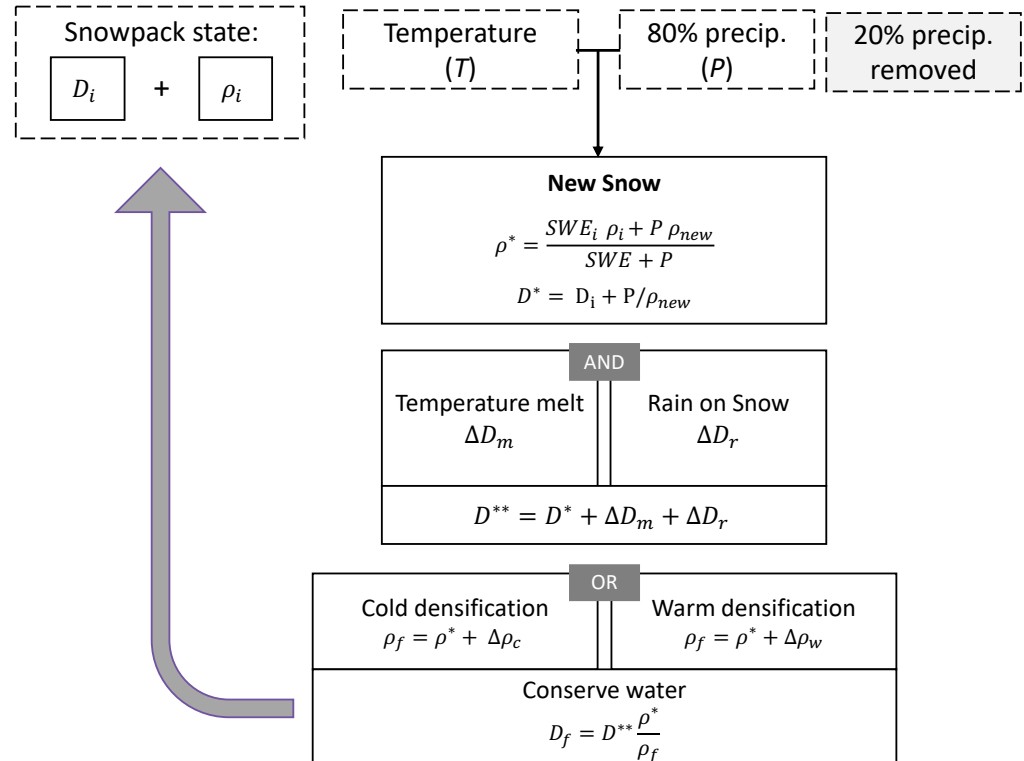

**Fig. 1 Conceptual overview of the Brown Temperature Index Model (B-TIM). At every time step and location, temperature and precipitation values are used to compute either the density and depth of any new snow or the**
**temperature of any rainfall. The snowpack state (snow depth and density) is affected by rain melt, melting due to air temperature, and one of two densification processes which cause both depth and density variables to evolve.**



**Table 1 Parameter values and units for model equations.**

| Symbol | Value | Units | Equation |
|---|---|---|---|
| A | 67.9 | kg m$^{-3}$ | (1) |
| B | 51.3 | kg m$^{-3}$ | (1) |
| C | 2.6 | K$^{-1}$ | (1) |
| $M_1$ | $4.08 \times 10^{-7}$ | m (snow) K$^{-1}$ hr$^{-1}$ | (3) |
| $M_2$ | $9.96 \times 10^{-5}$ | mm (w.e.) K$^{-1}$ hr$^{-1}$ | (3) |
| $T_{melt}$ | -1.0 | °C | (4) |
| $C_w$ | $4.18 \times 10^3$ | J kg$^{-1}$ K$^{-1}$ | (5) |
| $T_{freeze}$ | 0.0 | degrees C | (5) |
| $L_f$ | $0.334 \times 10^6$ | J kg$^{-1}$ | (5) |
| $\rho_{water}$ | 1000 | kg m$^{-3}$ | (5) |
| $C_1$ | 2.0 | m$^{-2}$ | (7a) |
| $C_2$ | 0.028 | m$^3$ kg$^{-1}$ | (7a) |
| $B_1$ | 0.6 | unitless | (7a) |
| $C_3$ | 0.08 | K$^{-1}$ | (7a) |
| $W_1$ | 204.70 | mm (w.e.) | (7b) |
| $W_2$ | 0.673 | m | (7b) |
| $W_{max}$ | 700 | kg m$^{-3}$ | (7b) |
| $\Delta t$ | 3600 | s | (7c) |
| $a$ | $2.778 \times 10^{-6}$ | s$^{-1}$ | (7c) |

*Initialization, meteorological driving, and time stepping*

Each simulated snow year initializes from snow-free conditions on August 1 and runs until the following July 31.
Two-metre temperature and total precipitation (frozen and solid) are the only inputs to the model; the specific
variables we used from each reanalysis are listed in Table 2. A fixed 20% precipitation reduction is implemented
at each model time step as a general loss parameter—this captures canopy interception, sublimation, and blowing
snow for frozen precipitation. The variable $P$ represents the reduced precipitation in metres of water for a given
time step and location. The model time step is one hour, but less frequent driving data can be handled. If needed,
the model linearly interpolates temperature to hourly steps and divides accumulated precipitation by the duration
of the driving data time step in hours.

**Table 2 Variables used from each reanalysis. "T" refers to the 2-meter temperature variable and "P" refers to the total**
**precipitation, both of which are used to drive the B-TIM. "SWE" refers to the snow water equivalent variable from**
**each reanalysis.**

|  | Reanalysis | Model Variable | Units | Frequency |
|---|---|---|---|---|
| T | ERA5/ERA5Snow | Parameter ID 167: "t2m" | K | 1h |
|  | JRA-55 | Parameter Code 11: "TMP" | K | 3h |



| | MERRA2 | inst1_2d_asm_Nx: "T2M" | K | 1h |
|---|---|---|---|---|
| P | ERA5/ERA5Snow | Parameter ID 288: "tp" | m/hr | 1h |
| | JRA-55 | Parameter Code 61: "TPRAT" | mm/day | 3h |
| | MERRA2 | "PRECTOTLAND" | mm/s | 1h |
| SWE | ERA5 | Parameter ID 141: "sd" | m (water equivalent) | 1h |
| | ERA5Snow | Available on request. | m (water equivalent) | 1h |
| | JRA-55 | Parameter Code 65: "SNWE" | mm (water equivalent) | 3h |
| | MERRA2 | "SNOMAS" | mm (water equivalent) | 1h |

*Determining precipitation phase*

At each model time step, the precipitation phase is classified as snow or rain using a 0°C threshold. Previous B-TIM applications allowed mixed precipitation between 0° and 2°C following a linear relationship. For large scale study, there is little advantage to including mixed precipitation according to the linear relationship as opposed to a fixed threshold. The absence of mixed precipitation has a minimal impact on the aggregated variables, though it causes local differences in regions with ephemeral snow.


*Updating snow depth and density*

Following Hedstrom & Pomeroy (1998), frozen precipitation during a time step is assigned a "new snow" density,

$$\rho_{new} = A + Be^{T/C}, \ T < 0°C, \tag{1}$$

where $T$ is the air temperature (values of the constants are listed in Table 1).


Intermediate values for snow depth and density are assigned to the model's single snow layer.

$$D^* = D_i + P\left(\frac{\rho_w}{\rho_{new}}\right) \tag{2a}$$

$$\rho^* = \frac{(D_i\rho_i + P\rho_w)}{D^*} \tag{2b}$$

Three densification/melting steps are then applied to evolve $\rho^*$ and $D^*$.

     1.   Snow melt is computed at each model time step using a melt factor, $\gamma$ (mm w.e. K$^{-1}$ hr$^{-1}$), which is based on the intermediate snow layer density, $\rho^*$. The relationship used to calculate $\gamma$ is based on Kuusisto (1984):

$$\gamma = M_1\rho^* - M_2. \tag{3}$$



Lower and upper bounds of $4.1 \times 10^{-3}$ and $0.23$, respectively, are enforced on $\gamma$. Hourly melt, represented as the change in snow depth $\Delta D_m$, follows a standard temperature index approach:

$$\Delta D_m = \begin{cases} -\frac{(T - T_{melt})}{\rho^*} \gamma, & T > T_{melt} \\ 0, & T \leq T_{melt} \end{cases} \tag{4}$$

where $T_{melt} = -1°C$ is the threshold air temperature used for snow melt.

The leading coefficient in Eq. 3, $M_1$, has been halved relative to and Brown et al. (2003) to reduce the rate of snow melt during the ablation season. This has been implemented for the CMC snow product.

2. Snow melt caused by rainfall on the snowpack is computed using

$$\Delta D_r = -\frac{R \rho_w C_w (T_w - T_{freeze})}{L_f \rho^*}. \tag{5}$$

$C_w$ is the heat capacity of water (J kg$^{-1}$ K$^{-1}$), $R$ is the total rainfall (m), $T_w$ is the rainfall temperature (°C), and $L_f$ is the latent heat of fusion for ice (J kg$^{-1}$). Rain temperature is taken to be equal to air temperature, as in Brown et al. (2003), and the snowpack is assumed to be isothermal and 0°C, implying instant melting of the snowpack when it is warmed.

3. A second intermediate snow depth is computed based on these first two steps.

$$D^{**} = D^* + \Delta D_m + \Delta D_r \tag{6}$$

Depending on air temperature, one of two possible densification processes is implemented. Both processes initially affect density (Equations 7a and 7b), and then snow depth is adjusted to conserve total water.

*Cold:* When temperatures are below $T_{melt}$, cold snow aging is implemented as follows:

$$\Delta \rho_D = C_1 (SWE^*) \exp[\, C_3 \, (T_{melt} - T_{snow}) \,] \exp[\, -C_2 \rho^* \,], \quad T < T_{melt} \tag{7a}$$

$SWE^*$ is the snow water equivalent (kg m$^{-2}$, calculated as the product $\rho^* D^{**}$). $C_1$ and $C_2$ are empirically derived constants. This formulation was proposed in Anderson (1976) and the parameters used in the B-TIM are in the accepted range. $T_{snow}$ is the snow temperature, taken to be equal to air temperature in this step. Snowpack density is allowed to vary between 200 and 550 kg m$^{-3}$. The densification process does not vary seasonally.

*Warm:* When temperatures are above -1°C, the snowpack undergoes a warm settling process, which increases the density more rapidly. A maximum density is first defined with dependence on the intermediate snow depth:

$$\rho_{max} = W_{max} - \frac{W_1}{D^{**}} \left( 1 - \exp \left[ -\frac{D^{**}}{W_2} \right] \right), \tag{7b}$$

and then adjusted by the intermediate density:

$$\Delta \rho_D = (\rho_{max} - \rho^*)(1 - e^{-a\Delta t}), \quad T \geq -1°C. \tag{7c}$$

The value of $a$ is such that in a one model time step ($\Delta t = 3600 \, s$), the density difference is adjusted by 1% of $(\rho_{max} - \rho^*)$, which constitutes a change in density of a few percent for typical values of $\rho^*$.

The final density is calculated as

$$\rho_f = \rho^* + \Delta \rho_D, \tag{8}$$

and the final depth is calculated after the densification process in the following manner to conserve water:

$$D_f = D^{**} \left( \frac{\rho^*}{\rho_f} \right). \tag{9}$$



The final snow depth and density are carried to the next model time step and new meteorological forcing is read in. The values of the prognostic variables are recorded at daily frequency and saved in monthly files. Annual total SWE and maximum SWE are tracked over the model year and single values for each are saved at the end of the run.

## 2.2 Reanalysis products

In this work, we use three current generation reanalyses which produce snow variables for 40 years or more for the Northern Hemisphere. We use the ECMWF Reanalysis, version 5, "ERA5" (Dutra et al., 2012; Hersbach et al., 2020), the second-generation Modern-Era Retrospective analysis for Research and Applications from the National Aeronautics and Space Administration, "MERRA-2" (Gelaro et al., 2017; Reichle et al., 2017), and the Japanese Meteorological Agency's 55-year Reanalysis, "JRA-55" (Kobayashi et al., 2015). These products differ from one another with respect to data assimilation schemes as well as their component atmospheric and land models (Table 3). All three global reanalyses assimilate conventional atmospheric measurements, but ERA5 and JRA-55 additionally assimilate snow depth observations and satellite derived snow extent information. The different techniques used to constrain ERA5 and JRA-55 SWE using snow cover observations are described below in more detail. Additional comparisons of the reanalyses are documented in the Supplementary Information Sect. 1.

Beginning in 2004, ERA5 assimilates the Interactive Multisensor Snow and Ice Mapping System (IMS) snow cover product wherever the model first guess indicates snow free conditions (de Rosnay et al., 2015). In the IMS snow cover product, grid cells are either snow covered or snow-free. Snow-free observations are treated as observations of 0 cm snow depth, while observations of full snow cover are treated as 5 cm of snow depth. These observations, together with the in situ snow depth measurements, enter the 2D-OI scheme to update the snow depth. The inclusion of IMS snow cover in the data stream reduces overall snow amounts and is associated with a discontinuity in ERA5 snow. We highlight this effect through comparison with ERA5Snow, a data product produced by an offline run of the ECMWF land model. ERA5Snow is produced with the same land surface data assimilation as ERA5 except the IMS satellite snow product [de Rosnay, private access to data]. It is separate from the offline ERA5-Land product produced by ECMWF.

JRA-55 constrains snow using passive microwave observations from 1987 to the present, and climatological snow cover fills gaps back to 1980. Though the microwave data processing methods are not fully documented in the peer reviewed literature, Kobayashi et al. (2015) say the estimates of snow cover extent come from comparing brightness temperature at different frequencies (37 GHz and 19 GHz at both horizontal and vertical polarization) to regionally and seasonally varying thresholds. All the snow is removed from grid cells where the land surface analysis indicates the presence of snow and the satellite observations do not. Snow is added to grid cells where the land surface analysis does not indicate snow but the satellite observations do. Unlike the fixed relationship between snow cover and snow depth used in ERA5, when the algorithm adds snow in JRA-55, it is a variable snow depth that would reduce land surface temperatures to freezing if it were to melt. Wherever the satellite and land surface analyses agree (both report snow covered conditions or both report no-snow conditions), no adjustment is made.



### 2.3 Temperature and precipitation biases

Biases in temperature and precipitation directly impact online and offline snow products, and our aim is to isolate and characterize their effects. Briefly comparing temperature and precipitation fields from the three reanalysis products, MERRA-2 exhibits the lowest hemispheric mean land temperatures for most of the year, and JRA-55 the highest (Fig. 2a). In the winter months, the JRA-55 mean temperature exceeds that of ERA5 by 2.15 K and MERRA2 by nearly 3 K, with the largest temperature difference occurring in January. In addition to being the coldest on average, MERRA-2 has the largest land area capable of sustaining snow, diagnosed as regions with $T < 0°C$ (Fig. 2b). This frozen land area exceeds that of ERA5 by 1 million km$^2$ or more during the shoulder seasons of autumn and spring.

With respect to total precipitation, JRA-55 is about 10% wetter than the other two products across all months. MERRA-2 and ERA5 agree more closely, with differences of just 1% in autumn and spring. ERA5 is 4% wetter in the winter, and MERRA-2 is about 6% wetter in the summer months. We investigate the roles of these forcing biases in SWE biases by implementing a simple climatological bias correction (method is described in SI Sect. 2).



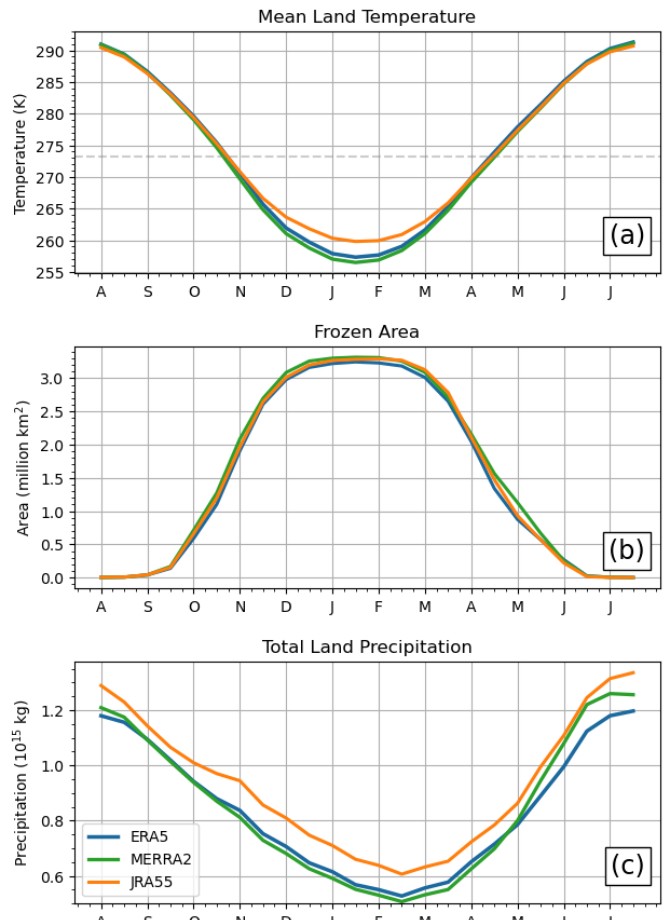

**Fig. 2 Climatologies of mean temperature, frozen area, and total precipitation over Northern Hemisphere land areas, computed twice monthly using 14-day windows centred on the 1$^{st}$ and 15$^{th}$ of each month. JRA-55 is wetter than the other datasets, and several degrees warmer in the winter months. Over the winter, MERRA-2 is the driest and coldest.**

### 2.4 Topography, land mask, and regional definitions

Mountain regions are excluded from our analysis using a mask derived from the Global Earth Topography and Sea Surface Elevation at 30 arc second resolution digital elevation model (GETASSE30 DEM). Locations in the DEM with local slope greater than 2° are defined as mountainous. After coarsening the slope mask (to 0.25°x0.25°, the ERA5 resolution), grid cells that are more than 95% mountainous are recorded in a binary mountain mask file which is coarsened as needed using a nearest neighbour algorithm (for the MERRA-2 or JRA-55 grids).

To define land grid cells, we use the land-sea masks associated with each reanalysis. The land fraction is used to scale grid cell land area when computing total snow mass, which depends on SWE and land area. When possible, computations are done on a dataset's native grid, and conservative regridding is applied to the SWE data, conserving total snow mass, before calculating grid-dependent metrics.



### 2.5 *In situ* validation of SWE datasets

We evaluate the SWE from (offline) B-TIM and (online) reanalysis are evaluated by comparing them to a
combined historical snow course and airborne gamma derived SWE dataset. These data are independent from
snow data assimilated in JRA-55 and ERA5. Snow course observations involve manual measurements of snow
depth and density along a predefined transect, with measurements averaged to obtain a single SWE value for each
transect on a specific date (WMO, 2018). The measurement frequency for snow courses varies by jurisdiction,
ranging from monthly measurements in Alaska, the western continental US, and most of Finland, to measurements
every five days during the spring snowmelt period in Russia. The Russian network has the highest sampling
frequency and is well-distributed across the landscape, while dense networks with lower sampling frequency are
found in Finland, the northeast US, and parts of southern Canada. Airborne gamma SWE estimates are calculated
by differencing snow-free and snow-covered measurements after accounting for background soil moisture. Flights
are 15-20km long with a 300m wide footprint. Data are available for the United States and southern parts of some
Canadian provinces. There is broad consistency between snow courses and airborne gamma observations
(Mortimer et al. in discussion), so we are confident in using both types of information together to evaluate the two
types of products.

Using the method in Mortimer et al. (in discussion), reference SWE data are matched in space and time with the
gridded product data. Data are then spatially aggregated and summarized using bias, unbiased root mean squared
error (uRMSE), and correlation. We compare data pairs for November through March for all years between 1980
and 2020, aiming to include as many measurements as possible before the snow melt period. The validation is
performed on non-mountainous points with non-zero SWE values below 500 mm that are simultaneously
available for the reference data and all the estimates. The latter condition excludes some snow courses in coastal
areas due to differing land/ice/water masks and is consistent with our snow mass calculations. For our results,
there is no spatial aggregation by land type.

## 3 Results

In this study, we compare snow from global reanalyses (ERA5, ERA5Snow, MERRA2, and JRA-55) to snow
from offline B-TIM runs. These are named BrE5, BrM2, and BrJ55, reflecting the use of distinct reanalysis
meteorology for each version, but the same B-TIM snow model. Standardizing through using a single model
means that differences between the offline B-TIM runs only reflect differences in the forcing data.

### 3.1 B-TIM compares well to *in situ* observations.

Comparing modeled snow to *in situ* observations is one way to assess the realism and performance of each
product. In general, we find a much broader spread in modeled SWE for high reference SWE values, showing
overall decreasing model skill with increasing snow depth (Figs. 3a-f). In some products (JRA-55 and MERRA-
2), there is a cluster of points where the modeled snow is shallow, but reference SWE indicates deep snow. The
B-TIM products have greater absolute bias than their respective reanalysis products, but they are of comparable
magnitude. When mountain points are included, the B-TIM products have lower absolute bias than the reanalyses
(not shown). Of the reanalyses, ERA5 and ERA5Snow have the lowest uRMSE and highest correlation compared



to the reference values, so they outperform JRA-55 and MERRA-2 overall. By these measures, the three B-TIM products have comparable skill to ERA5/ERA5Snow. Notably, unlike their reanalysis counterparts, BrJ55 and BrM2 do not display the cluster of false low snow values.

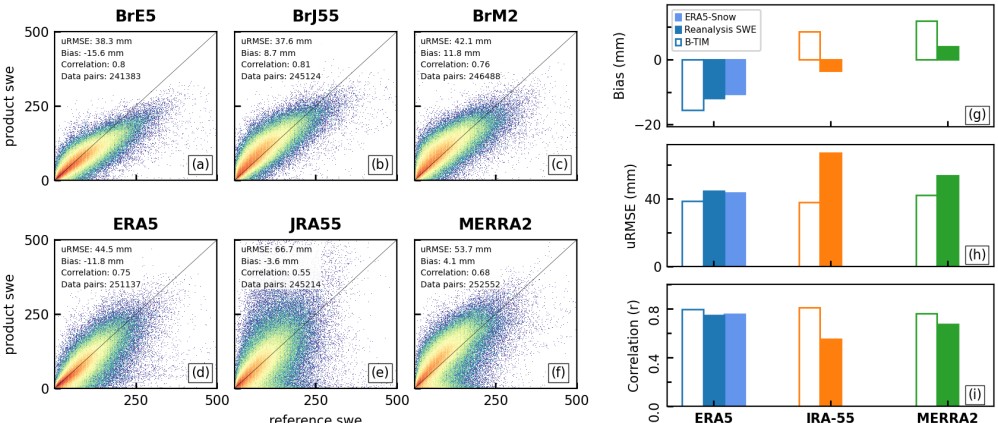

**Fig. 3 SWE product validation against snow course and gamma SWE measurements. Figs. 3a-f consist of scatterplots**
**showing all valid data pairs (snow course, product) from November to March over 1980-2018. Summary statistics, including the bias, unbiased root mean squared error (uRMSE), and correlation, are included in the legend and are summarized in Figs. 3g-i. In all cases, the reconstructed datasets have lower uRMSE and higher correlation than do the reanalysis datasets. uRMSE is calculated by removing the mean from the reference SWE and each set of product SWE values, then calculating the root mean squared error with those unbiased datasets.**

These validation results show two things. First, the offline products capture realistic snow patterns when compared to ground measurements, even in the context of snow from more complex coupled reanalyses. Second, we see that snow data assimilation does not guarantee skilful snow modeling by these measures. In particular, ERA5/ERA5Snow and JRA-55 are both produced with snow cover data assimilation (see Section 2.2), but while the former two are the best performing products, the latter performs poorly (with high uRMSE and low correlation)

and struggles both with false low snow values and large overestimates relative to ground truth. MERRA-2 does not assimilate snow data but also performs moderately by the comparison metrics. Additionally, model complexity does not guarantee skilful snow modelling. The offline products generated with the B-TIM, with neither snow data assimilation nor coupled interactions between snow and the land-atmosphere system, perform comparably to each other and to ERA5/ERA5Snow, despite differences in the forcing data.

**3.2 Using B-TIM to assess discrepancies between reanalysis snow products**

**3.2.1 Discrepancies in reanalysis snow climatologies are caused by forcing data biases**

Marked differences appear in the magnitude of total snow mass and snow covered area for the products considered here. Among the online (the B-TIM) datasets, JRA-55 (BrJ55) has the highest peak snow mass, exceeding the maximum value from MERRA-2 (BrM2) by about 0.15 x $10^{15}$ kg (0.17 x $10^{15}$ kg) and the maximum value from

ERA5 (BrE5) by 0.73 x $10^{15}$ kg (0.77 x $10^{15}$ kg), as seen in Fig. 4. The relative ranking of these products and the biases in the peak snow mass are closely reproduced by the offline model; since the offline model can reproduce the biases, we explore the possibility that they are directly caused by the forcing biases discussed in Sect. 2.3, which can equally affect both types of products. We test if these inter-product biases can be manipulated – in



particular, minimized – by bias correcting the meteorological fields used to drive the B-TIM. As described in the
SI, this correction is done by applying monthly correction factors to the forcing data computed for each pair of
datasets and each variable (temperature, precipitation). Then, for each possible pair (e.g. ERA5 with target
MERRA-2), three experiments are run: one with adjusted temperature, one with adjusted precipitation, and one
with both variables adjusted. This yields 18 datasets in addition to the three unadjusted B-TIM datasets, the three
reanalysis datasets, and ERA5Snow.

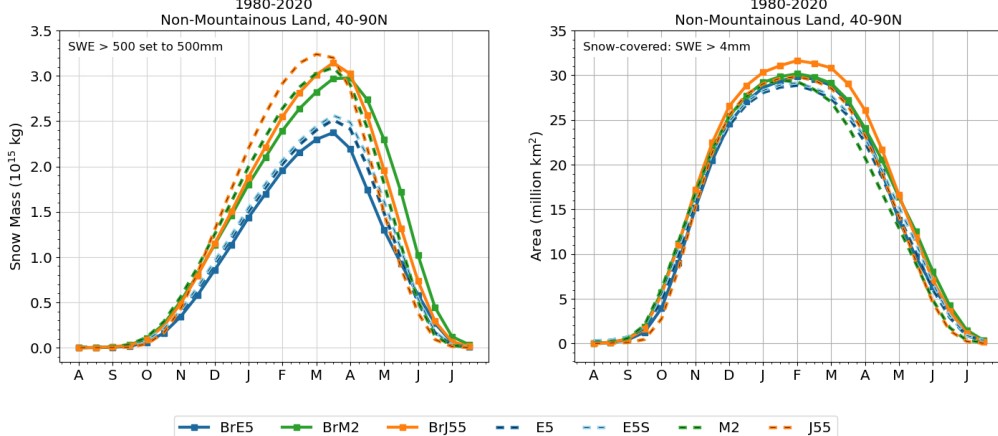

**Fig. 4 (a) Snow mass climatology over Northern Hemisphere land, with grid cells exceeding 500 mm capped at 500 mm.
4 (b) Snow-covered area climatology, calculated using areas of grid cells with more than 4 mm SWE.**

Comparing the bias-adjusted versions of BrE5 and BrM2 (Fig. 5) indicates that temperature biases are the main
driver for the differences between ERA5 and MERRA-2 snow mass and snow cover shown in Fig. 4; in the
experiments where the ERA5 and MERRA-2 temperature climatologies are bias adjusted, the resulting snow
fields are also much more similar. Precipitation biases play a smaller role and correcting the precipitation
modestly decreases the snow mass biases over the whole season. Snow covered area is not very sensitive to the
precipitation correction, though the best agreement in both cases comes from rescaling both variables. However,
mean biases in forcing variables do not explain all the difference in SWE. For the pairs involving JRA-55 (Figs.
S1 and S2), the precipitation correction improves the agreement between a dataset and a chosen target, but not at
the level observed for the ERA5-MERRA-2 pair; the temperature scaling sometimes degrades the agreement.
However, JRA-55 is several degrees warmer and about 10% wetter than the other two reanalyses on average over
the region of interest (Figure 2), constituting more substantial differences than exist between ERA5 and MERRA-
2.



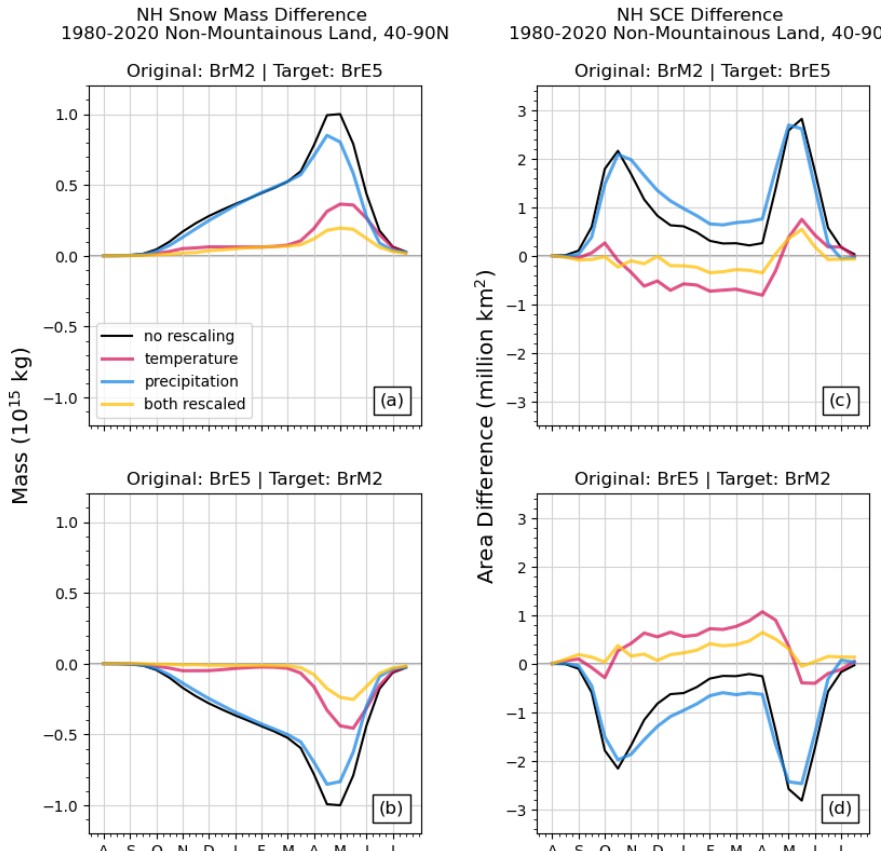

**Fig. 5 (a, b) NH snow mass and (c, d) snow cover extent differences for MERRA-2 and ERA5 calculated as *original* minus *target*. Each panel shows the difference between the original and target snow mass climatologies (black) and the coloured lines represent the datasets generated by adjusting temperature (pink), precipitation (blue), or both (yellow) to the target dataset's climatology.**


To summarize, the B-TIM products (BrE5, BrM2, BrJ55) retain the relative biases present in the reanalyses. Motivated by this, we have explored the potential use of bias correction on the meteorological forcing to elucidate the drivers of these snow biases or to correct them to first order. This approach isolates a subset of drivers and gives insight into the dominant sources of snow biases, but the approach requires more refinement to explain

biases (see discussion in Sect. 4) more fully.

### 3.2.2 B-TIM versions of SWE fields show consistency in seasonal cycle, interannual variability.

Aside from JRA-55, which has delayed snow accumulation but an early peak SWE, all the other datasets agree that the snow mass maximum occurs within a two-week period centred on March 15. For snow covered area, all datasets except MERRA-2 peak during the 14-day period centred on Feb 1; the MERRA-2 maximum occurs two

weeks earlier. Thus, unlike the reanalyses, the B-TIM products provide more consistent descriptions of key snowpack climatology metrics.



Figure 6 shows the September-October-November (SON) mean snow mass time series, calculated over land regions from 40-90N (excluding mountains), with the same 500 mm maximum imposed as before to exclude high

SWE values over isolated grid cells. Figure 7 shows the December-January-February (DJF) time series of mean snow mass. In these figures, dashed lines are used for reanalysis snow, and the solid lines show offline snow. Even without detrending and removing the mean, it is clear that the solid lines are highly consistent with each other (for both continents and both seasons; panels a and c), while there is much more disagreement between reanalysis products. This highlights the role that factors other than forcing biases play in introducing inter-product

differences. We quantify the consistency in the offline-offline and reanalysis-reanalysis pairs by calculating correlation coefficients after removing the least-squares linear fit (Fig. 8). Across all regions and all seasons, the B-TIM products are strongly correlated with one another (r > 0.85), whereas the reanalysis r-values are lower in general and greatly depend on the pair.


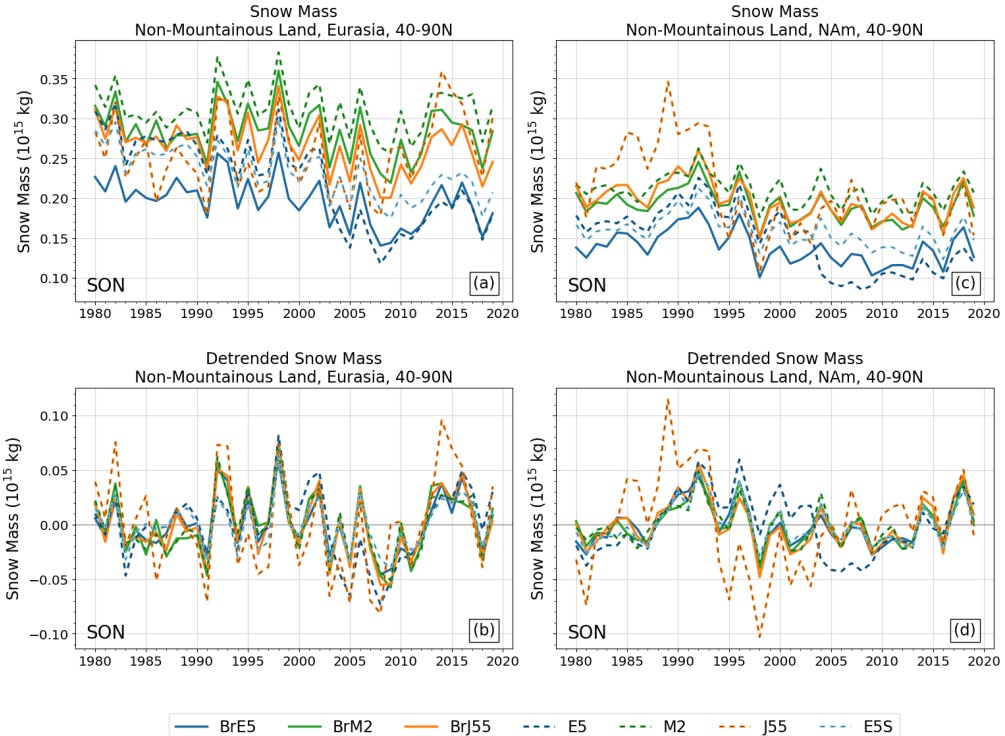

**Fig. 6 Time series of total snow mass for SON by continent. Lower row has linear trends removed. All datasets but the native ERA5 and JRA-55 show similar interannual variability, including ERA5Snow. JRA-55 differences are greatest over North America.**



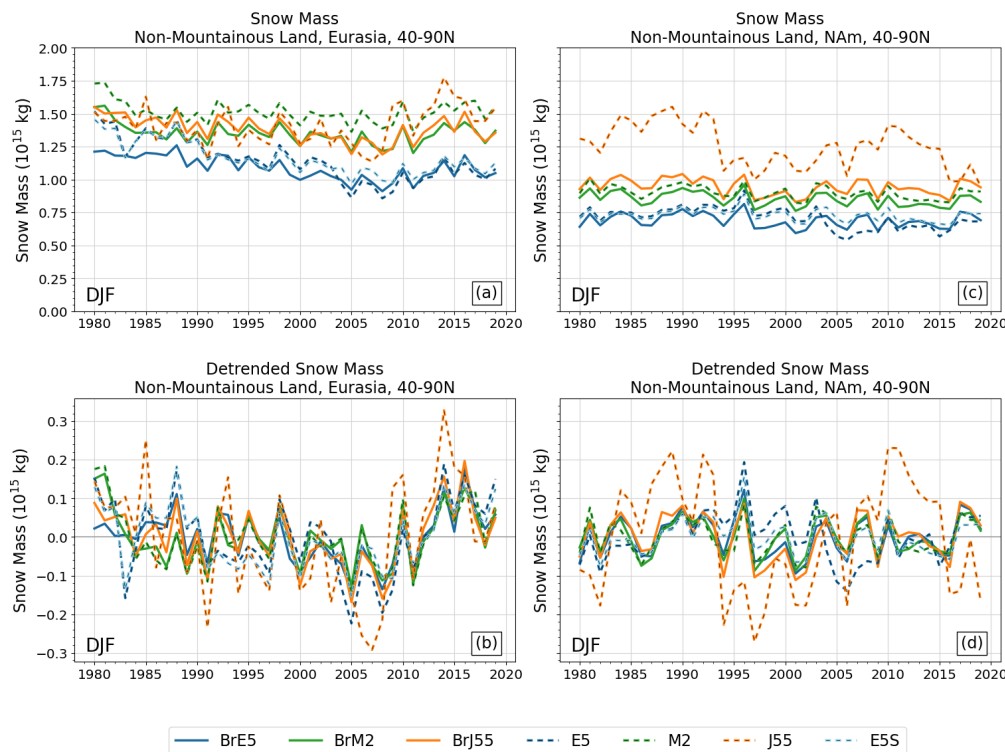

**Fig. 7 Time series of total snow mass for DJF by continent. Lower row has linear trends removed. All datasets but the native ERA5 and JRA-55 show similar interannual variability, including ERA5Snow. JRA-55 differences are greatest over North America.**

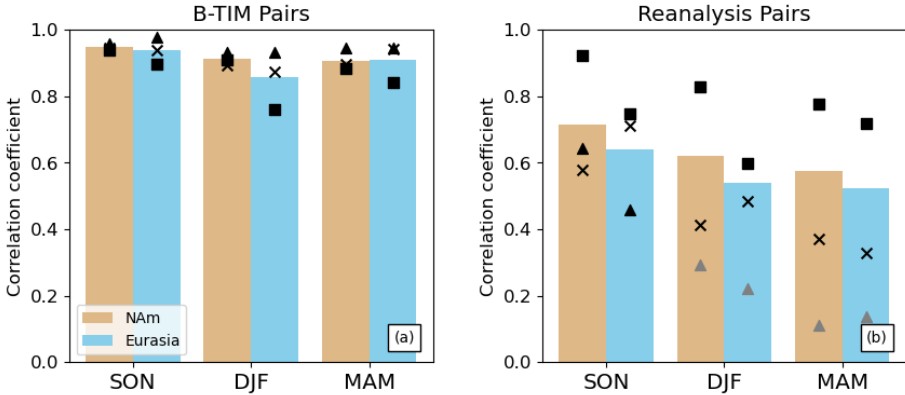

**Fig. 8 (a) Correlation coefficients for B-TIM dataset pairs. Individual values are shown with black points, and the mean is represented by the height of the bar to summarize the group. Similar information is shown in Fig. 8 (b) for the reanalysis dataset pairs. The values are much lower across all seasons and both regions for native datasets, indicating a higher degree of disagreement. Generally, agreement degrades over the course of the snow season. The ERA5 and JRA-55 pair is represented by the triangle, ERA5 and MERRA-2 by the square, and JRA55 and MERRA-2 with the x.**



The reanalysis JRA-55 snow mass is unique, characterized by large decadal variations. Positive anomalies are most common from 1980-1994 and 2010-2020, while negative anomalies occur from 1995-2009 (Fig. 7d). These inconsistencies are not as extreme over Eurasia, as JRA-55 captures positive and negative anomalies that are mostly in agreement with the remaining datasets, but its variations have the greatest magnitude (e.g., 1991, 2014).

The disagreement is substantial in terms of snow mass amount. Over North America, especially before 1995, the reanalysis JRA-55 dataset has as much as 50% more snow mass than the other reanalyses. This behaviour is not present in BrJ55.

We now return to consider the two versions of the ERA5 reanalysis: ERA5 and ERA5Snow (dashed, blue in two

shades, Figs. 6 and 7). The two timeseries diverge due to a change to the snow cover extent data assimilation in 2004. The mean difference in DJF snow mass over North America between these two products is five times greater after 2004 compared to before 2004 ($9 \times 10^{13}$ kg and $1.8 \times 10^{13}$ kg, respectively) and three times greater after 2004 for Eurasia ($7.7 \times 10^{13}$ kg compared with $2.4 \times 10^{13}$ kg). This step change is problematic for trend and correlation assessments, so we use ERA5Snow in Figure 9 below. As an offline product, BrE5 does not display

the step change in 2004.

These two examples show that B-TIM snow datasets can generate benchmark datasets that are not sensitive to complexity that is added through data assimilation or other factors. The comparison between reanalysis and the offline product forced with the same meteorology can highlight spurious variability, as in the case of JRA-55, or

point to temporal inhomogeneities, as with ERA5.

The consistency found for the offline products extends to spatial patterns. The time series of the DJF spatial pattern correlation between dataset pairs is shown in Fig. 9, with SON values shown in Figure S3. For both seasons, offline-offline pairs are the most consistent with each other (with the highest r-values), despite different

meteorological forcing. There is also evidence of spatial disagreement between some of the reanalysis products. Notably, JRA-55 is very different from all the other datasets. This can be seen by the ERA5Snow-JRA-55 and MERRA-2-JRA-55 pairs (different model, different forcing), which have the weakest spatial correlations, or by the BrJ55-JRA-55 pair (different model, same forcing), which has a much lower correlation compared to the other same-forcing pairs. These pattern correlations appear stable across the 40-year period for all pairs, although those

involving JRA-55 have larger year-to-year variability.





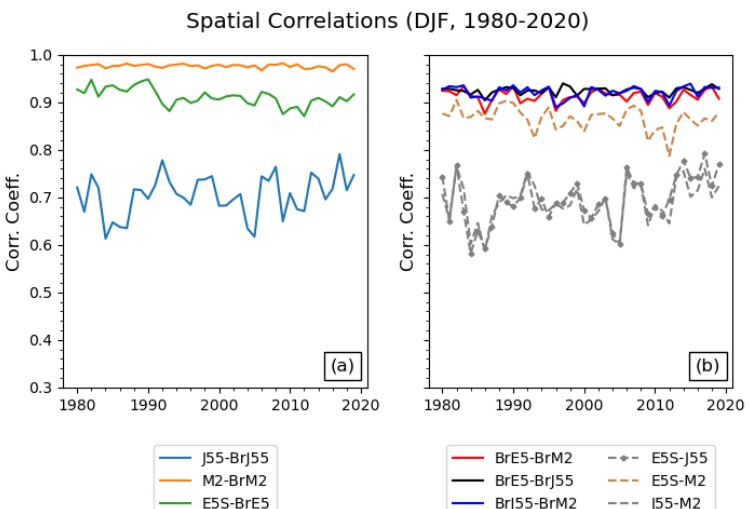

**Fig. 9 Spatial correlations for DJF calculated between pairs of datasets with the same meteorology (a) and between pairs of similar type (b; either offline-offline or reanalysis-reanalysis). Comparisons involving JRA-55 have the lowest spatial correlations in all cases.**

### 4 Discussion and conclusions

To summarize our key points:

• An updated and more complete description of the B-TIM offline snow model has been provided for the first time since 2003, accompanied by an open-source code release of the model implemented in Python.

• Offline B-TIM snow generated using meteorological forcing from three reanalysis products validates well against an independent set of *in situ* snow observations (Sect. 2.5). The offline products perform generally as well as (online coupled) reanalysis snow. Based on this result, datasets generated with the

B-TIM are treated as reasonably performing benchmark estimates of historical snow and used to investigate discrepancies in reanalysis SWE.

• Compared to online reanalysis snow, offline B-TIM snow yields far more consistent interannual variability both for aggregate and spatially resolved snow metrics. This suggests the potential utility of the B-TIM as an offline tool for simplified snow modelling in seasonal to decadal prediction systems

and climate downscaling for impacts analysis.

• Climatological characteristics of offline B-TIM snow are generally more consistent with one another for various measures than reanalysis snow, despite differences in the meteorological forcing data. Using B-TIM with bias-adjusted forcing, climatological SWE differences between ERA5 and MERRA2 are found to primarily come from temperature biases (MERRA2 is colder, resulting in more SWE throughout the

snow season). Attribution of discrepancies of JRA-55 with the other two reanalyses is not as straightforward, as we discuss next.





Offline modelling has allowed us to understand some of the components contributing to the spread in SWE estimates across these three reanalyses. In general, nonlinearities inherent to snow modeling mean that it is unclear
how exactly meteorological biases will impact modeled SWE fields both for historical and modeled future snow conditions (Evan & Eisenman, 2021; Räisänen, 2023; Sospedra-Alfonso & Merryfield, 2017). Interpreting the causes of SWE differences is further complicated when comparing products produced not only using different snow models, but also different data assimilation schemes. In this sense, the B-TIM can easily generate simplified reference datasets (no data assimilation, and a single, simple model) alongside more complex products of interest.
Here, we have attempted to attribute climatological SWE biases to climatological meteorological biases by adjusting each of the two forcing variables and calculating the effect on the SWE. We have taken advantage of B-TIM's speed, which has allowed us to perform many cross-tests.

The simple bias adjustment methodology we use requires more refinement to fully explain the biases, as the JRA-
55 comparisons are not as clear as the comparison between ERA5 and MERRA-2. Large differences between a dataset and a chosen target may make the multiplicative scaling less suitable. For example, it could substantially change the shape of the distribution of precipitation intensity. Additionally, other aspects of the driving variables can influence SWE in models, such as the diurnal cycle in temperature and the distribution of precipitation intensity/duration (especially near freezing temperatures, when precipitation may change phase between rain and
snow). Multiplicative rescaling can affect these features when matching a dataset to a chosen target, with the greatest impact coming from adjusting both driving variables at once. These effects are most relevant at the shoulder seasons and for areas with ephemeral snow.

We have shown that terrestrial SWE taken directly from the JRA-55 reanalysis is problematic and should not be
used for climate analysis. Unlike the BrJ55 product, which performs comparably to the BrE5 and BrM2 products, the reanalysis JRA-55 terrestrial snow product is the least accurate with respect to the *in situ* validation. Furthermore, the interannual variability of the JRA-55 snow mass anomaly time series (Figs. 5 and 6) and corresponding SWE field patterns (Fig. 7) differ greatly from the other datasets. These results suggest that the problem with JRA-55 snow arises from the snow data assimilation.


Snow is a critical component of the climate system, influencing a range of environmental and societal processes. Accurate snow modeling is needed for applications that require a long time series (e.g. trend analysis) and the best instantaneous estimates of SWE (e.g. numerical weather prediction). We have here demonstrated the value of a simple model like B-TIM to help benchmark new products as they are developed. These considerations will
continue to be important as we look ahead to the next generation of global reanalyses, including the JMA Reanalysis for Three Quarters of a Century (JRA-3Q; Kobayashi et al., 2021) and ERA6 (Balsamo et al., 2023).

**Code and Data Availability**

ERA5 data were retrieved from the Copernicus Climate Data Store (single levels: https://doi-org.myaccess.library.utoronto.ca/10.24381/cds.adbb2d47). ERA5-Snow data are available on request from
patricia.rosnay@ecmwf.int. JRA-55 data were retrieved from the NCAR Research Data Archive (all collections: https://rda.ucar.edu/datasets/ds628.0/dataaccess/). MERRA-2 data were retrieved from the Goddard Space Flight



Center Distributed Active Archive Center (GSFC DAAC). The combined snow reference dataset will become available by the time of publication.

Processed output from the B-TIM runs and reanalysis data, required to reproduce the figures, are archived at https://doi.org/10.5683/SP3/IV6SVJ.

**Author Contribution**

AEC produced the updated B-TIM code and generated the modeled snow data. In-situ validation was done by CM with input from LM and CD. The snow data comparison method was developed jointly by AEC, PJK, LM,

and CD and executed by AEC. AEC prepared the manuscript with contributions from all co-authors.

**Competing Interests**

Some authors are members of the editorial board of journal The Cryosphere. The authors have also no other competing interests to declare.

**Acknowledgements**

The results contain modified Copernicus Climate Change Service information. Neither the European Commission nor ECMWF is responsible for any use that may be made of the Copernicus information or data it contains.

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
