# Peer review of "A simple snow temperature index model exposes discrepancies between reanalysis snow water equivalent products"

_EGUsphere, 2024_

## Author Comment (AC1)

**Responses to Reviewer #1**

*This manuscript presents how a simple off-line snow temperature index model (B-TIM) can be considered to highlight discrepancies between snow water equivalent (SWE) products from three reanalysis (JRA-55, ERA5 and MERRA-2) and an additional product (ERA5Snow) for historical period (1980-2020). The authors used either biased, or adjusted temperature and precipitation from reanalysis as input data for B-TIM. The SWEs produced with B-TIM and various sets of input data were then compared to the SWEs produced by the reanalysis. Climatological characteristics and interannual variability were investigated. To carry out this study, they improved and translated the previous version of B-TIM and made it publicly available. This manuscript opens up the possibility of using a simple off-line model for large-scale snow cover studies.*

We thank the reviewer for all the comments and have included responses along with requested additional calculations in the Author Comments. Line references included here refer to the original manuscript.

*Some modifications in the structure could lighten the text and focus more on the results and the contribution of using a simple off-line model like B-TIM. For example, a large part of section 2.1 would have a more appropriate place in the SI. After all, this is not a paper about improving B-TIM, but more about using it. If this is not the case, please change the title and specify this aspect more clearly in the objectives.*

The primary objective of the model description section is to thoroughly document the settings of the B-TIM before analyzing its output and using it. When model decisions are not defined explicitly, it can be challenging for others to reproduce those decisions and/or compare across datasets fairly. To this end, we propose to move Tables 1 and 2 to the SI while maintaining the written text describing the time evolution of the modeled snow. This way, the section will be shortened without breaking up the model description.

*It would be useful to remind the reader of the context in the results. For example, simply add a sentence to remind us that ERA5 and ERA5Snow have the same meteorology, which explains why we don't have BrE5S.*

Thank you for the suggestion to clarify this point. The following text will be added after L270:
Two of the reanalysis datasets, ERA5 and ERA5Snow, share the same meteorology. Therefore, there is only one BrE5 dataset that is produced.

*Methodological choices (e.g., bias adjustment) could be justified in greater detail, with more references where possible.*

The methodological choices stem from the hypothesis that major differences in SWE across reanalyses are consequences of the mean bias in the temperature and precipitation, as is currently described in L308-309. This motivation could be strengthened in the bias adjustment section.

We propose an addition around L309: If biases in mean meteorological conditions are the primary source of snow bias, a correction toward similar climatological conditions should yield more similar modeled snow. We implement a first-order correction which relies on monthly correction values derived from climatological temperature and precipitation conditions (see SI).

In the literature, correction factors similar to ours are typically derived from in-situ observations for the purpose of model calibration, validation, or sensitivity testing (e.g. Cho et al., 2022 or Raleigh et al., 2015). Our application is different in the sense that we are adjusting the forcing data in both directions between pairs of datasets to investigate whether we can bridge the gap between them and then attribute the differences to the mean biases. Some of this additional context will be added to the discussion.

*The results focus on very large domains (Northern Hemisphere, Eurasia, North America). It would have been interesting to look at these results more regionally.*

Regional studies are of major interest, especially with respect to a breakdown according to land type. The purpose of this paper is to characterize snow over simplified terrain and at large scales, but the two accompanying papers Mudryk et al. (in discussion) and Mortimer et al. (in discussion) provide some regional results and perhaps should be more clearly highlighted here.

We propose to use the following to replace the text starting in L77: This study has as its focus hemispheric snow. Even at these large scales and excluding the complicated case of mountain snow, there are discrepancies that should be characterized. For exploration into regional performance, there are two other studies prepared for publication, Mudryk et al. (in discussion) and Mortimer et al. (in discussion), which include all the datasets discussed in this paper. Mudryk et al. (in discussion) compares a suite of 23 snow datasets, ranks the performance of each one, and examines the inter-dataset consistency. Mortimer et al. (in discussion) presents an expanded SWE dataset that combines in-situ and airborne SWE measurements and assesses snow dataset performance when compared to the in-situ record. This in-situ data is used across all three studies. Our main scientific work…

*Temperature and precipitation bias were corrected with a multiplicative factor. As precipitation is a zero-bound variable, it is generally corrected by a multiplicative method, whereas temperature is often corrected by an additive method. The choice of this method is not sufficiently justified. Can you explain thoroughly why you chose a multiplicative factor and why you apply it this way?*

The methodology requires some more explanation, which we outline here and will endeavour to integrate into the paper. Commonly, when correction factors are based on the bias between ground-truth and modeled data, normally distributed biases are corrected with an additive method, and lognormally distributed biases are corrected by multiplicative adjustment. This usually means temperature biases are corrected by adding a constant, while precipitation biases are scaled multiplicatively. This has the added benefit of maintaining the zero-bound of precipitation. The application here is different. We use the multiplicative factor for both variables because:

- Climatological *temperature* biases between datasets are small (no more than 2 % of the absolute temperature) year-round. For fractionally small differences compared to the raw value, a multiplicative method and an additive method yield very similar results.*
- We did some filtering to ensure that no erroneously large or small corrections were applied (e.g. from division by a small precipitation value)
  - Precipitation scaling factors were limited between 1/3 and 3, and wherever dry conditions prevailed in either the native or target dataset (daily average precip. <0.2 mm), a scaling factor of 1 was used (no adjustment).
  - Temperature scaling factors were limited between 0.99 and 1.01
- Using the same method simplified the experiment runs

*Explicit comparison of the two methods:

Given Dataset 1 (D1) and target Dataset 2 (D2), both of which are functions of location and time, let climatological temperature conditions for a given month be represented by C1 and C2. With a multiplicative bias correction as in the paper, the temperature at time step $t$ is adjusted as in (1).

$$D1_{mult}(t) = D1(t)\frac{C2}{C1} = D1(t) + D1(t)\left(\frac{C2}{C1} - 1\right) = D1(t) + D1(t)\frac{(C2-C1)}{C1} \tag{1}$$

Meanwhile, an additive method corrects D1 following (2).

$$D1_{add}(t) = D1(t) + (C2 - C1) = D1(t) + D1(t)\frac{(C2-C1)}{D1(t)} \tag{2}$$

The correction terms differ because D1(t) and C1 are not identical. However, both methods yield the right climatology, $\overline{D1_{add}} = \overline{D1_{mult}} = C2$.

If we write D1 in terms of departures from the climatology C1, we can rewrite the scaling factor in equation (2). The two scaling factors are close as long as the sub-daily variations $\delta D1$ are small compared to C1.

$$\frac{(C2-C1)}{D1(t)} = \frac{C2-C1}{C1+\delta D1(t)} = \frac{C2-C1}{C1\left(1+\frac{\delta D1(t)}{C1}\right)} \approx \frac{C2-C1}{C1}\left(1 - \frac{\delta D1(t)}{C1}\right) \tag{3}$$

Though the scaling factors are similar, multiplication can stretch or compress the diurnal cycle -- this can theoretically affect snow accumulation when temperatures are near freezing (primarily near the margins) but it should be a reasonably small effect. Given the reasoning above, we do not expect a large difference between the correction methods, and a preliminary test has been run to support this.

*Figure captions contain results, whereas captions should only contain descriptions of the elements present in the figure (colors, symbols, etc.). Please remove the result part in the captions.*

The captions of Figs. 2, 3, 6, 7, 8, 9 have been edited to remove results. They were all already described in the text.

*L102: Did you perform tests regarding the 20% of precipitation reduction?*

The 20% precipitation reduction is a value based on estimates of snow loss that were incorporated and tested during model development (Brown, 2003 and personal communication). The original validation was based on optimizing agreement with in-situ data at locations in eastern Canada. Given some recent increases in the availability and quality of in situ SWE, snow depth, and snow density information, future B-TIM development may revisit this 20% reduction. However, it is outside the scope of our current study and we did not test it.

Example citation for new validation data: Vionnet et al., 2021 or Mortimer et al. (in discussion).

*L109: Table 2. Please find a more consistent way to present column "Model variable". For example: « t2m (ID 167) » instead of « Parameter ID 167: "t2m" ». Add the model variable name for SWE in ERA5Snow. Also, this table could go in the SI, as it doesn't provide much relevant information to the text.*

Thank you for the suggestions. The intention was to copy the descriptions as directly as possible from the modeling centers, but a standardized approach is likely clearer. We will update Table 2 and move it to the SI, with reference to it in L102.

*Table 2, L215, L268, L428 : Modify MERRA2 to MERRA-2.*

Thank you for identifying these, they have all been fixed.

*L232: To take advantage of the fact that you have an SI, it might be interesting to present the differences in domains used for the different reanalyses (land grid points, mountainous grid points, etc.).*

It is an interesting suggestion to look at different domains. We hope the new pointers to the evaluation papers Mudryk et al., (in discussion) and Mortimer et al., (in discussion) will be sufficient for interested readers to turn to for regional analysis. It is best to interpret the strengths and limitations of these products in a larger ensemble.

*Fig. 3 : Please describe colors used in the scatterplots in the caption; modify ERA5-Snow to ERA5Snow; consider using hatches for ERA5Snow in the right panel and present the legend in a neutral color.*

We have improved the clarity of Fig. 3 with consistent labeling and both colour and hatching to distinguish ERA5Snow and ERA5.

**Please note the following change: the values presented in this figure are slightly different from the previous version (no changes to discussion/conclusions). This resulted from mistakenly using Nov-May data to produce the figure in the original manuscript. This version correctly uses Nov-March data.**

[Figure]

**Fig. 3 SWE product validation against snow course and gamma SWE measurements. Figs. 3a-f consist of scatterplots showing all valid data pairs (snow course, product) from November to March over 1980-2018. Summary statistics, including the bias, unbiased root mean squared error (uRMSE), and correlation, are included in the legend and are summarized in Figs. 3g-i. uRMSE is calculated by removing the mean from the reference SWE and each set of product SWE values, then calculating the RMSE with those datasets.**

In L293 we will include the following description: The scatterplots (figs. a-f) are coloured as heatmaps to display the concentration of points, which is highest where the colour is red. Each scatterplot represents over 200 000 pairs of points.

*L374: Modify JRA55 to JRA-55.*

Fixed.

*L469 : Modify ERA5-Snow to ERA5Snow.*

Fixed.

---

## Author Comment (AC2)

**Responses to Reviewer #2**

*I echo the summary and comments of the first Reviewer, so I will not repeat them here. This is a solid analysis, but there are some points that should be addressed.*

*General comment:*

*The goal of this work (from my understanding) is to create benchmark snow datasets using an offline model, which is potentially more consistent than reanalysis or coupled model snow products that can be affected by uncertainties and errors related to forcing, data assimilation, model bias, and coupling. The authors assert that offline modeling can "isolate the role of meteorological driving" from these other issues. This is largely true. However, I would caution the authors that the model they are using still has a number of parameters whose values they chose, and which affect the snow output from the model. With a different set of parameters, the model could (or arguably will) provide a different indication of the amount of error introduced by meteorological forcing, because the model dynamics will change. So, it's not entirely possible to disentangle forcing uncertainty from model construction and parameter uncertainty, without an exhaustive analysis of model sensitivity. I would recommend that the authors qualify their statements by noting that this is only one parameter set for this model, and their findings might be different if the parameters in Table 1 were changed.*

Thank you for the thoughtful consideration of this paper and the context in which we discussed the results. Studies of snow mass/SWE uncertainty are most frequently done by comparing (fixed versions of) various snow models, including output from reanalyses. Sometimes, this precludes investigation of forcing biases and structural biases caused by model choices.

As noted, analyzing a model's sensitivity to parameter (or process) changes can be and has been done systematically in some cases (e.g. Essery, 2015 or Raleigh et al., 2015). While it is outside of the scope of this study to develop the B-TIM, recent increases in the availability and quality of in situ SWE, snow depth, and snow density information may feed into future development. We will incorporate discussion of this structural/parameter uncertainty into the manuscript.

"Isolat[ing] the role of meteorological driving" is meant in the sense that the offline inter-product differences are not a function of snow modeling differences, while the online inter-product differences are. These products may still be biased relative to ground-truth, which is explored in Fig. 3 and the new supplementary figure AC Fig. 3 included below.

*Specific comments:*

*Lines 66-70: Past studies have attempted to assess the influence of various factors on snow model uncertainty, including forcing, and it would be appropriate to cite one or more here (e.g., Raleigh et al., 2015, https://doi.org/10.5194/hess-19-3153-2015).*

Thank you for this idea. The Raleigh et al., 2015 study and others (e.g. Cho et al., 2022, Essery, 2015, Günther et al., 2019, and Menard et al., 2021) have explored these various factors. They find that snow modeling is sensitive to forcing biases and parameter changes, and provide

avenues to assessing these sensitivities. We will include some reference to this work around L442. Our method offers a way to decompose or investigate snow biases when it is impossible to run additional simulations or directly interact with the model. This is the case for reanalysis snow, as in this study.

*Fig. 1: The 20% of precipitation loss seems quite arbitrary. I realize that this constant derives from the Brown et al. 2003 paper, but there is no reason to assume that this loss rate would be consistent across sites. This parameter could have a strong influence on the magnitude of snow accumulation. Can the authors give some indication of why a constant 20% is the best choice?*

This is an important open question. The 20% reduction does derive from previous studies and tuning that was done with respect to in-situ snow data at a limited number of sites. For some time, a varying loss parameter was used in the B-TIM for different snow classes (following Sturm et al., 2010). The 20% reduction was applied to tundra, prairie, and taiga snow-climate zones as the most likely regions where blowing snow and sublimation could dominate. This was simplified by extending the same reduction to the rest of the NH land, as these are the snow-climate zones that take up most of the Northern Hemisphere. This practice has continued due to robust performance of the modeled snow, but regional performance is almost certainly affected.

While it is outside the scope of the current study, the spatial variability of and sensitivity to this precipitation loss factor have not been recently characterized. Given recent increases in the availability and quality of in situ SWE, snow depth, and snow density information, future B-TIM development should revisit this 20% reduction.

*Fig. 1: What are delta rho_c and delta rho_w? I do not see them mentioned anywhere else in the text?*

Thank you for catching this omission. These two variables represent the change in snowpack density under "cold" and "warm" compaction processes. Equations 7a, 7c, and 8 will be corrected.

$$\Delta\rho_c = C_1 \, (\text{SWE}^*) \exp[\, C_3 \, (T_{melt} - T_{snow}) \,] \exp[\, -C_2\rho^* \,], \ T < T_{melt} \tag{7a}$$

$$\Delta\rho_w = (\rho_{max} - \rho^*)(1 - e^{-a\Delta t}), \ T \geq -1°C. \tag{7c}$$

$$\rho_f = \rho^* + \Delta\rho_w \text{ if } T \geq -1°C \text{ else } \rho_f = \rho^* + \Delta\rho_c, \tag{8}$$

*Lines 279-280: The authors state that ERA5 outperforms JRA-55 and MERRA-2, based on uRMSE and correlation. However, Figure 3g seems to show that the bias is higher for ERA5. Shouldn't the bias be important here as well? What about raw RMSE (without removing the bias)? I would guess that most users of these datasets are unlikely to unbias them before using them.*

The bias is important, but because it is calculated as the sum of differences, positive and negative differences can cancel and yield a small bias. The RMSE measures the average magnitude of the error (weighting larger errors more heavily), so it avoids the cancellation problem. However,

RMSE and bias are not independent pieces of information, as any bias that exists contributes to the RMSE. That is why we report uRMSE instead.

$$RMSE = \sqrt{uRMSE^2 + bias^2}$$ (1)

Generally, the products considered here have small bias compared to uRMSE, as provided in the table below. Each B-TIM product has a lower RMSE than its reanalysis counterpart and the greatest RMSE arises from the JRA-55 online snow product. A comment about this will be incorporated in the manuscript to supplement Fig. 2.

|  | BIAS | URMSE | RMSE |
|---|---|---|---|
| **BRE5** | -11 | 32 | 34 |
| **BRJ55** | 10 | 33 | 34 |
| **BRM2** | 8 | 36 | 37 |
| **ERA5** | -9 | 38 | 39 |
| **JRA55** | 4 | 61 | 61 |
| **MERRA2** | 9 | 46 | 47 |

*Line 345: The authors find that the "B-TIM products provide more consistent descriptions of key snowpack climatology metrics". This is true, but consistency does not necessarily mean accuracy. It's possible that one of the reanalyses is a more accurate reflection of reality. The authors could use their in-situ data to evaluate this, but have not yet sufficiently done so in this manuscript.*

Thank you for requesting further clarification of these statements. We have done some additional assessments with the in-situ data to supplement these claims (AC Fig. 3).

*Line 366: Why did the authors not use a more robust trend method, like Theil-Sen slope (which is less influenced than OLS by outliers, and the start and end of time series), for detrending?*

The Theil-Sen estimator gives a robust linear regression. As was noted, it is less influenced by outliers and the start/end of time series than the OLS minimization method. To the authors' knowledge, Theil-Sen slope is well defined, but there are several definitions of the y-intercept in the literature. The definition of y-intercept that we used to produce the figure below is the one implemented in the scipy stats Python module.

$$median(y) - TheilSenSlope \times median(x)$$ (2)

Regardless of detrending method, the same qualitative results are seen. All the B-TIM datasets display the same variability and diverge notably from JRA-55 (throughout 1980-2020) and ERA5 (after 2004).

We propose to include: Detrending by another method yields similar results (e.g. using the Theil-Sen estimator, which is robust to outliers and shifts to the start or end of the time series, not shown).

[Figure]

**AC Fig. 1: Same as Fig. 7 in the original manuscript, but detrending in panels (b) and (d) was done based on Theil-Sen line fitting.**

*Fig. 8: Interesting to make point that differences among reanalysis are greater than among B-TIM. Isn't that kind of expected? Wouldn't it also be informative to compare B-TIM vs. reanalysis pairs (same forcing, different models), using more than just correlation (as in Fig. 9, but using bar charts as in figure 8, for example)?*

On one hand, it is reasonable to expect that differences among reanalyses are greater than among B-TIM datasets. However, model differences could theoretically be introducing snow biases of opposing sign (e.g. one model with too much melt and another with too little melt could increase or reduce the bias in the snow depending on the overall bias). Therefore, it is important to document the finding in this case.

The suggestion to look at same-forcing pairs is a good one. Spatial correlation is one aspect of their agreement that we show in Fig. 9, and rather than relying on the visual comparison (i.e. looking at the same-colour lines on Figs. 6 and 7), we can include a figure in the supplementary

information to address this question. A preliminary version of this figure is below (AC Fig. 2). The pair with JRA forcing (orange) is poorly correlated across all seasons and both continents. It is consistently worse over Eurasia for a given season, and the correlation drops over the snow season. The other pairs have higher correlations, with the MERRA pair being most similar. This lines up with the discussion of Fig. 9 that is based on spatial correlations.

[Figure]

**AC Fig. 2: Correlation between SON, DJF, and MAM snow mass time series with the same forcing. BrE5-ERA5 pair in blue, BrM2-MERRA2 pair in green, and BrJ55-JRA55 pair in orange. Values are split for two continents.**

*Lines 422-425 (section 4, 3rd bullet): The authors show that the B-TIM model results in "far more consistent interannual variability" than the reanalysis products. However, this does not necessarily mean that the B-TIM interannual variability is more "correct" (i.e. a more accurate representation of the true interannual variability). Can the authors show using their in-situ data that less (or more consistent) interannual variability results in greater accuracy?*

Discussion below.

*Lines 457-459: Similar comment as above. The authors suggest that there is a "problem" with JRA-55. This is a strong statement to make. It's true that JRA is the least accurate by some metrics and different from the other reanalyses, so it's possible that the authors' suggestion is correct. However, the authors have not shown in this manuscript that the interannual variability of JRA-55 is wrong. Maybe the interannual variability in the other reanalyses is too muted? The authors have in-situ data available to back up their statement, but they have not yet sufficiently done so in the manuscript.*

Thank you for the suggestion to fold the in-situ data into the discussion of JRA-55. We have produced some additional analysis covering Dec-Feb that indicates poor performance of JRA-55. AC Fig. 3 indicates that the version of JRA-55 that has less interannual variability (BrJ55, solid orange) also has significantly lower RMSE and higher correlation with the in-situ data than the native JRA-55 (dashed orange). All the products with similar interannual variability have lower RMSE and high correlation with in-situ data.

Finally, further evidence can be found in Mudryk et al. (in discussion). In that comparison of 23 snow products, JRA-55 is consistently among the lowest-ranking of them. This analysis breaks down mountainous, Arctic, and midlatitude regions. A discussion of this can also be included in the discussion.

[Figure]

**AC Fig. 3: Time series of validation metrics. DJF measurements only. Dashed lines are native SWE (ERA5, MERRA2, and JRA-55), while solid are B-TIM outputs (BrE5, BrM2, BrJ55).**

---

## Author Response (AR1)

**Responses to Reviewer #1**

*This manuscript presents how a simple off-line snow temperature index model (B-TIM) can be considered to highlight discrepancies between snow water equivalent (SWE) products from three reanalysis (JRA-55, ERA5 and MERRA-2) and an additional product (ERA5Snow) for historical period (1980-2020). The authors used either biased, or adjusted temperature and precipitation from reanalysis as input data for B-TIM. The SWEs produced with B-TIM and various sets of input data were then compared to the SWEs produced by the reanalysis. Climatological characteristics and interannual variability were investigated. To carry out this study, they improved and translated the previous version of B-TIM and made it publicly available. This manuscript opens up the possibility of using a simple off-line model for large-scale snow cover studies.*

We thank the reviewer for the helpful suggestions and overall supportive comments. The author responses to each point are included here. Line references refer to the revised manuscript.

*Some modifications in the structure could lighten the text and focus more on the results and the contribution of using a simple off-line model like B-TIM. For example, a large part of section 2.1 would have a more appropriate place in the SI. After all, this is not a paper about improving B-TIM, but more about using it. If this is not the case, please change the title and specify this aspect more clearly in the objectives.*

The primary objective of the model description section is to thoroughly document the settings of the B-TIM before analyzing its output and using it. When model decisions are not defined explicitly, it can be challenging for others to reproduce those decisions and/or compare across datasets fairly. We moved Tables 1 and 2 to the SI while maintaining the written text describing the time evolution of the modeled snow. This way, the section is shortened without breaking up the model description.

*It would be useful to remind the reader of the context in the results. For example, simply add a sentence to remind us that ERA5 and ERA5Snow have the same meteorology, which explains why we don't have BrE5S.*

Thank you for the suggestion to clarify this point; it is a good reminder. The following text has been added to L279: Two of the reanalysis datasets, ERA5 and ERA5Snow, share the same temperature and precipitation inputs ("meteorology"). Therefore, there is only one BrE5 dataset that is produced.

*Methodological choices (e.g., bias adjustment) could be justified in greater detail, with more references where possible.*

The methodological choices stem from the hypothesis that major differences in SWE across reanalyses are consequences of the mean bias in the temperature and precipitation, as was previously described in L currently described in L364-365. We have made edits to strengthen this motivation.

We have added the following to L365-369: If biases in mean meteorological conditions are the primary source of snow bias, a correction toward similar climatological conditions should yield more similar modeled snow. We implement a first-order correction which relies on monthly correction values derived from climatological temperature and precipitation conditions (see SI).

In the literature, correction factors similar to ours are typically derived from in-situ observations for the purpose of model calibration, validation, or sensitivity testing (e.g. Cho et al., 2022 or Raleigh et al., 2015), however these are not our objectives.

This is now discussed in L73-85: In this work, we use a fixed version of the B-TIM without further calibration or tuning. Therefore, one parameter set for the model is used and the results may still contain model bias. Quantifying this bias for the B-TIM model can be done thorough analysis of parameter and error sensitivity (Essery, 2015; Raleigh et al., 2015). However, our aim is to investigate reanalysis snow biases; each offline snow product will have the same model bias, whereas the coupled reanalysis snow does not. Comparing offline snow products therefore narrows down the sources of discrepancy without requiring a re-run of the complex snow modeling and data assimilation process.

*The results focus on very large domains (Northern Hemisphere, Eurasia, North America). It would have been interesting to look at these results more regionally.*

Regional studies are of major interest, especially with respect to a breakdown according to land type. The purpose of this paper is to characterize snow over simplified terrain and at large scales, but the two accompanying papers Mudryk et al. (in discussion) and Mortimer et al. (in discussion) provide some regional results and perhaps should be more clearly highlighted here.

Between L85-91, we added: This study has as its focus hemispheric snow. Even at these large scales and excluding the complicated case of mountain snow, there are discrepancies that should be characterized. For exploration into regional performance, there are two other studies prepared for publication, Mudryk et al. (in discussion) and Mortimer et al. (in discussion), which include all the datasets discussed in this paper. Mudryk et al. (in discussion) compares a suite of 23 snow datasets, ranks the performance of each one, and examines the inter-dataset consistency. Mortimer et al. (in discussion) presents an expanded SWE dataset that combines in-situ and airborne SWE measurements and assesses snow dataset performance when compared to the in-situ record. This in-situ data is used across all three studies. Our main scientific work…

*Temperature and precipitation bias were corrected with a multiplicative factor. As precipitation is a zero-bound variable, it is generally corrected by a multiplicative method, whereas temperature is often corrected by an additive method. The choice of this method is not sufficiently justified. Can you explain thoroughly why you chose a multiplicative factor and why you apply it this way?*

The methodology requires some more explanation and we thank the reviewer for the opportunity to add more information.

SI L40: Scaling factor values are bounded by 0.33 and 3 for precipitation and 0.99 and 1.01 for temperature.

SI L42: Commonly, when correction factors are based on the bias between ground-truth and modeled data, normally distributed biases are corrected with an additive method, and lognormally distributed biases are corrected by multiplicative adjustment. This usually means temperature biases are corrected by adding a constant, while precipitation biases are scaled multiplicatively. This has the added benefit of maintaining the zero bound of precipitation. Given the small climatological temperature biases (no more than 2% of absolute temperature), multiplicative and additive methods yield similar results. We used a multiplicative method for both variables to simplify the experiment runs.

L533-539: Large differences between a dataset and a chosen target may make the multiplicative scaling less suitable; for example, by changing the input variable distributions significantly. Additionally, other aspects that are not captured in mean conditions can influence SWE in models, such as the nature of the diurnal cycle in temperature and the distribution of precipitation intensity/duration (or a combination of the two).

*Explicit comparison of the two methods:

Given Dataset 1 (D1) and target Dataset 2 (D2), both of which are functions of location and time, let climatological temperature conditions for a given month be represented by C1 and C2. With a multiplicative bias correction as in the paper, the temperature at time step $t$ is adjusted as in (1).

$$D1_{mult}(t) = D1(t)\frac{C2}{C1} = D1(t) + D1(t)\left(\frac{C2}{C1} - 1\right) = D1(t) + D1(t)\frac{(C2-C1)}{C1} \qquad (1)$$

Meanwhile, an additive method corrects D1 following (2).

$$D1_{add}(t) = D1(t) + (C2 - C1) = D1(t) + D1(t)\frac{(C2-C1)}{D1(t)} \qquad (2)$$

The correction terms differ because D1(t) and C1 are not identical. However, both methods yield the right climatology, $\overline{D1_{add}} = \overline{D1_{mult}} = C2$.

If we write D1 in terms of departures from the climatology C1, we can rewrite the scaling factor in equation (2). The two scaling factors are close as long as the sub-daily variations $\delta D1$ are small compared to C1.

$$\frac{(C2-C1)}{D1(t)} = \frac{C2-C1}{C1+\delta D1(t)} = \frac{C2-C1}{C1\left(1+\frac{\delta D1(t)}{C1}\right)} \approx \frac{C2-C1}{C1}\left(1 - \frac{\delta D1(t)}{C1}\right) \qquad (3)$$

*Figure captions contain results, whereas captions should only contain descriptions of the elements present in the figure (colors, symbols, etc.). Please remove the result part in the captions.*

The captions of Figs. 2, 3, 6, 7, 8, 9 have been edited to remove results. They were all already described in the text.

*L102: Did you perform tests regarding the 20% of precipitation reduction?*

This question came up in both reviews, and we agree that model development should continue to be a priority. Given some recent increases in the availability and quality of in situ SWE, snow depth, and snow density information, future B-TIM development may revisit this 20% reduction. However, it is outside the scope of our current study, and we did not test it.

524-531: Future work should continue developing the B-TIM through systematic testing of parameter values. For example, the spatial variability of and sensitivity of the model to the 20% precipitation loss have not recently been characterized. This type of work is possible due to the recent increases in the availability and quality of in situ SWE, snow depth, and snow density information (Vionnet et al., 2021) for validation. Forcing biases and parameter changes both strongly influence modeled snow (Cho et al., 2022; Essery, 2015; Günther et al., 2019; Menard et al., 2021), and they should be characterized for the B-TIM. However, offline modeling can broadly be seen as a tool to investigate snow biases in products where additional simulations are not feasible, as is the case for reanalysis.

*L109: Table 2. Please find a more consistent way to present column "Model variable". For example: « t2m (ID 167) » instead of « Parameter ID 167: "t2m" ». Add the model variable name for SWE in ERA5Snow. Also, this table could go in the SI, as it doesn't provide much relevant information to the text.*

Thank you for the suggestions. The intention was to copy the descriptions as directly as possible from the modeling centers, but a standardized approach is likely clearer. We updated Table 2 and moved it to the SI, with reference to it in L123.

*Table 2, L215, L268, L428 : Modify MERRA2 to MERRA-2.*

Thank you for identifying these, they have all been fixed.

*L232: To take advantage of the fact that you have an SI, it might be interesting to present the differences in domains used for the different reanalyses (land grid points, mountainous grid points, etc.).*

It is an interesting suggestion to look at different domains. We hope the new pointers to the evaluation papers Mudryk et al., (in discussion) and Mortimer et al., (in discussion) will be sufficient for interested readers to turn to for regional analysis. It is best to interpret the strengths and limitations of these products in a larger ensemble. (L85-91, as described above)

*Fig. 3 : Please describe colors used in the scatterplots in the caption; modify ERA5-Snow to ERA5Snow; consider using hatches for ERA5Snow in the right panel and present the legend in a neutral color.*

We have improved the clarity of Fig. 3 with consistent labeling and both colour and hatching to distinguish ERA5Snow and ERA5.

**Please note the following change: the values presented in this figure are slightly different from the previous version (no changes to discussion/conclusions). This resulted from mistakenly using Nov-May data to produce the figure in the original manuscript. This version correctly uses Nov-March data.**

[Figure]

**Fig. 3 SWE product validation against snow course and gamma SWE measurements. Figs. 3a-f consist of scatterplots showing all valid data pairs (snow course, product) from November to March over 1980-2018. Summary statistics, including the bias, unbiased root mean squared error (uRMSE), and correlation, are included in the legend and are summarized in Figs. 3g-i. uRMSE is calculated by removing the mean from the reference SWE and each set of product SWE values, then calculating the RMSE with those datasets.**

In L316-318: The scatterplots (figs. a-f) are coloured as heatmaps to display the concentration of points, which is highest where the colour is red. Each scatterplot represents over 200 000 pairs of points.

*L374: Modify JRA55 to JRA-55.*

Fixed.

*L469 : Modify ERA5-Snow to ERA5Snow.*

Fixed.

**Responses to Reviewer #2**

*I echo the summary and comments of the first Reviewer, so I will not repeat them here. This is a solid analysis, but there are some points that should be addressed.*

*General comment:*

*The goal of this work (from my understanding) is to create benchmark snow datasets using an offline model, which is potentially more consistent than reanalysis or coupled model snow products that can be affected by uncertainties and errors related to forcing, data assimilation, model bias, and coupling. The authors assert that offline modeling can "isolate the role of meteorological driving" from these other issues. This is largely true. However, I would caution the authors that the model they are using still has a number of parameters whose values they chose, and which affect the snow output from the model. With a different set of parameters, the model could (or arguably will) provide a different indication of the amount of error introduced by meteorological forcing, because the model dynamics will change. So, it's not entirely possible to disentangle forcing uncertainty from model construction and parameter uncertainty, without an exhaustive analysis of model sensitivity. I would recommend that the authors qualify their statements by noting that this is only one parameter set for this model, and their findings might be different if the parameters in Table 1 were changed.*

We thank the reviewer for the thoughtful consideration of this paper and the context in which we discussed the results.

Studies of snow mass/SWE uncertainty are most frequently done by comparing (fixed versions of) various snow models, including output from reanalyses. Sometimes, this precludes investigation of forcing biases and structural biases caused by model choices. As noted, analyzing a model's sensitivity to parameter (or process) changes can be and has been done systematically in some cases (e.g. Essery, 2015 or Raleigh et al., 2015). While it is outside of the scope of this study to develop the B-TIM, recent increases in the availability and quality of in situ SWE, snow depth, and snow density information may feed into future development. We incorporated discussion of this structural/parameter uncertainty into the manuscript.

L72-79: In this work, we use a fixed version of the B-TIM without further calibration or tuning. Therefore, one parameter set for the model is used and the results may still contain model bias. Quantifying this bias for the B-TIM model can be done thorough analysis of parameter and error sensitivity (Essery, 2015; Raleigh et al., 2015). However, our aim is to investigate reanalysis snow biases; each offline snow product will have the same model bias, whereas the coupled reanalysis snow does not. Comparing offline snow products therefore narrows down the sources of discrepancy without requiring a re-run of the complex snow modeling and data assimilation process.

"Isolat[ing] the role of meteorological driving" is meant in the sense that the offline inter-product differences are not a function of snow modeling differences, while the online inter-product differences are. These products may still be biased relative to ground-truth, which is explored in Fig. 3 and the new figure SI Fig. 3.

*Specific comments:*

*Lines 66-70: Past studies have attempted to assess the influence of various factors on snow model uncertainty, including forcing, and it would be appropriate to cite one or more here (e.g., Raleigh et al., 2015, https://doi.org/10.5194/hess-19-3153-2015).*

Thank you for this idea. The Raleigh et al., 2015 study and others (e.g. Cho et al., 2022, Essery, 2015, Günther et al., 2019, and Menard et al., 2021) have explored these various factors. They find that snow modeling is sensitive to forcing biases and parameter changes, and provide avenues to assessing these sensitivities. Incorporated in the changes to L72-79 that were described above.

*Fig. 1: The 20% of precipitation loss seems quite arbitrary. I realize that this constant derives from the Brown et al. 2003 paper, but there is no reason to assume that this loss rate would be consistent across sites. This parameter could have a strong influence on the magnitude of snow accumulation. Can the authors give some indication of why a constant 20% is the best choice?*

This is an important open question. The 20% reduction does derive from previous studies and tuning that was done with respect to in-situ snow data at a limited number of sites. For some time, a varying loss parameter was used in the B-TIM for different snow classes (following Sturm et al., 2010). The 20% reduction was applied to tundra, prairie, and taiga snow-climate zones as the most likely regions where blowing snow and sublimation could dominate. This was simplified by extending the same reduction to the rest of the NH land, as these are the snow-climate zones that take up most of the Northern Hemisphere. This practice has continued due to robust performance of the modeled snow, but regional performance is almost certainly affected.

While it is outside the scope of the current study, the spatial variability of and sensitivity to this precipitation loss factor have not been recently characterized.

524-531: Future work should continue developing the B-TIM through systematic testing of parameter values. For example, the spatial variability of and sensitivity of the model to the 20% precipitation loss have not recently been characterized. This type of work is possible due to the recent increases in the availability and quality of in situ SWE, snow depth, and snow density information (Vionnet et al., 2021) for validation. Forcing biases and parameter changes both strongly influence modeled snow (Cho et al., 2022; Essery, 2015; Günther et al., 2019; Menard et al., 2021), and they should be characterized for the B-TIM. However, offline modeling can broadly be seen as a tool to investigate snow biases in products where additional simulations are not feasible, as is the case for reanalysis.

*Fig. 1: What are delta rho_c and delta rho_w? I do not see them mentioned anywhere else in the text?*

Thank you for catching this omission. These two variables represent the change in snowpack density under "cold" and "warm" compaction processes. Equations 7a, 7c, and 8 are corrected.

$$\Delta\rho_c = C_1 \, (\text{SWE}^*) \, \exp[\, C_3 \, (T_{melt} - T_{snow}) \,] \, \exp[\, -C_2\rho^* \,] \, , \; T < T_{melt} \qquad\qquad (7a)$$

$$\Delta\rho_w = (\rho_{max} - \rho^*)(1 - e^{-a\Delta t}), \ T \geq -1°C. \tag{7c}$$

$$\rho_f = \rho^* + \Delta\rho_w \text{ if } T \geq -1°C \text{ else } \rho_f = \rho^* + \Delta\rho_c, \tag{8}$$

*Lines 279-280: The authors state that ERA5 outperforms JRA-55 and MERRA-2, based on uRMSE and correlation. However, Figure 3g seems to show that the bias is higher for ERA5. Shouldn't the bias be important here as well? What about raw RMSE (without removing the bias)? I would guess that most users of these datasets are unlikely to unbias them before using them.*

The bias is important, but because it is calculated as the sum of differences, positive and negative differences can cancel and yield a small bias. The RMSE measures the average magnitude of the error (weighting larger errors more heavily), so it avoids the cancellation problem. However, RMSE and bias are not independent pieces of information, as any bias that exists contributes to the RMSE. That is why we report uRMSE instead.

$$RMSE = \sqrt{uRMSE^2 + bias^2} \tag{1}$$

Generally, the products considered here have small bias compared to uRMSE, as provided in the table below.

|  | BIAS | URMSE | RMSE |
|---|---|---|---|
| **BRE5** | -11 | 32 | 34 |
| **BRJ55** | 10 | 33 | 34 |
| **BRM2** | 8 | 36 | 37 |
| **ERA5** | -9 | 38 | 39 |
| **JRA55** | 4 | 61 | 61 |
| **MERRA2** | 9 | 46 | 47 |

L326-327: Low bias does not necessarily mean good performance, as individual positive and negative differences can cancel.

L329-330: The RMSE (calculated as the bias and uRMSE added in quadrature) of each offline snow product is less than that of its reanalysis counterpart.

L330-331: Each B-TIM product has a lower RMSE than its reanalysis counterpart and the greatest RMSE arises from the JRA-55 online snow product.

*Line 345: The authors find that the "B-TIM products provide more consistent descriptions of key snowpack climatology metrics". This is true, but consistency does not necessarily mean accuracy. It's possible that one of the reanalyses is a more accurate reflection of reality. The authors could use their in-situ data to evaluate this, but have not yet sufficiently done so in this manuscript.*

Thank you for requesting further clarification of these statements. We have done some additional assessments with the in-situ data to supplement these claims (SI Fig. 3).

*Line 366: Why did the authors not use a more robust trend method, like Theil-Sen slope (which is less influenced than OLS by outliers, and the start and end of time series), for detrending?*

The Theil-Sen estimator gives a robust linear regression. As was noted, it is less influenced by outliers and the start/end of time series than the OLS minimization method. To the authors' knowledge, Theil-Sen slope is well defined, but there are several definitions of the y-intercept in the literature. The definition of y-intercept that we used to produce the figure below is the one implemented in the scipy stats Python module.

$$median(y) - TheilSenSlope \times median(x) \tag{2}$$

Regardless of detrending method, the same qualitative results are seen. All the B-TIM datasets display the same variability and diverge notably from JRA-55 (throughout 1980-2020) and ERA5 (after 2004).

L421-423: Detrending by another method yields similar results (e.g. using the Theil-Sen estimator, which is robust to outliers and shifts to the start or end of the time series, not shown).

[Figure]

**AC Fig. 1: Same as Fig. 7 in the manuscript, but detrending in panels (b) and (d) was done based on Theil-Sen line fitting.**

*Fig. 8: Interesting to make point that differences among reanalysis are greater than among B-TIM. Isn't that kind of expected? Wouldn't it also be informative to compare B-TIM vs. reanalysis pairs (same forcing, different models), using more than just correlation (as in Fig. 9, but using bar charts as in figure 8, for example)?*

On one hand, it is reasonable to expect that differences among reanalyses are greater than among B-TIM datasets. However, model differences could theoretically be introducing snow biases of opposing sign (e.g. one model with too much melt and another with too little melt could increase or reduce the bias in the snow depending on the overall bias). Therefore, it is important to document the finding.

The suggestion to look at same-forcing pairs is great. Spatial correlation is one aspect of their agreement that we show in Fig. 9, and rather than relying on the visual comparison (i.e. looking at the same-colour lines on Figs. 6 and 7), we included a figure in the supplementary information to address this question (SI Fig. 5).

L479-482: Additionally, reanalysis and offline versions of JRA-55 snow have low spatial correlation across all seasons and both continents compared to the ERA5 and MERRA-2 (Fig.

S5). Broadly, reanalysis and offline patterns are less similar over Eurasia for a given season, and the correlation decreases over the year.

*Lines 422-425 (section 4, 3ʳᵈ bullet): The authors show that the B-TIM model results in "far more consistent interannual variability" than the reanalysis products. However, this does not necessarily mean that the B-TIM interannual variability is more "correct" (i.e. a more accurate representation of the true interannual variability). Can the authors show using their in-situ data that less (or more consistent) interannual variability results in greater accuracy?*

Discussion below.

*Lines 457-459: Similar comment as above. The authors suggest that there is a "problem" with JRA-55. This is a strong statement to make. It's true that JRA is the least accurate by some metrics and different from the other reanalyses, so it's possible that the authors' suggestion is correct. However, the authors have not shown in this manuscript that the interannual variability of JRA-55 is wrong. Maybe the interannual variability in the other reanalyses is too muted? The authors have in-situ data available to back up their statement, but they have not yet sufficiently done so in the manuscript.*

Thank you for the suggestion to fold the in-situ data into the discussion of JRA-55. We have produced some additional analysis covering Dec-Feb that indicates poor performance of JRA-55 compared to other datasets. SI Fig. 3 indicates that the version of JRA-55 that has less interannual variability (BrJ55, solid orange) also has significantly lower RMSE and higher correlation with the in-situ data than the native JRA-55 (dashed orange). All the products with similar interannual variability have lower RMSE and high correlation with in-situ data.

Finally, further evidence can be found in Mudryk et al. (in discussion). In that comparison of 23 snow products, JRA-55 is consistently among the lowest-ranking of them. This analysis breaks down mountainous, Arctic, and midlatitude regions.

L454-456: Additional comparison with *in situ* data indicates that the version of JRA-55 that has less interannual variability (BrJ55, solid orange) also has significantly lower RMSE and higher correlation with the in-situ data than the native JRA-55 (dashed orange; SI Fig. 3).

L547-L550: Complementary methods such as multi-model intercomparison also indicate poor performance for JRA-55. Across mountainous, Arctic, and midlatitude regions, it is consistently among the lowest-ranking products, while BrJ55 performs well (Mudryk et al., *in discussion*).

---

## Author Response (AR2)

**Responses and Minor Revisions**

*I would like to thank the authors for their satisfactory responses to my first round of revision. By including the comments and suggestions of the two reviewers, they produced a clearer and more complete manuscript.*

*During my reading of the revised version of the manuscript, I noted some minor points to address, mainly on the figures, which, once amended, will enable the manuscript to be improved even further. They are listed below.*

We thank the reviewer for the revisions suggested; we have cleared up some minor points relating to the figures and think that the clarity overall is improved. The changes are outlined below and can be found in the tracked changes version of the manuscript.

*L132 - Modify reference to Table 1 by Table S1.*
Done.

*L392 - Modify reference to Fig. 3 by Fig. S3.*
Done.

*Fig. 3 - Add the information relative to "The scatterplots are coloured as heatmaps to display the concentration of data points, which is highest where the colour is red. " in the caption.*
This clarification has been moved to the caption of Fig. 3.

*Fig. 8 - You have gray and black triangles. Is there a reason? If so, please explain; either plot all triangles in black.*
The gray points indicate that there was no statistically significant correlation between the time series in that pair (at 95% confidence level). This distinction has been added to the legend for clarity, along with the shapes used for the individual pairs.

*Fig. 9 (as well as Fig. S4) - For greater clarity, the colors used in the right-hand panel could be identical for each pair (E5S-J55 for example), with a difference in line style when the pair is of a different type (solid for offline-offline, dotted for reanalysis-reanalysis for example).*
This convention has been adopted for Figs. 9 and S4 for clarity.

*As suggested in the previous review, you might remind the reader that ERA5 and ERA5Snow have the same "meteorology", and that BrE5 is therefore used as the offline version with ERA5Snow.*
We have added a reminder (L423-425) to assist interpretation of these comparisons.

*Table S1 - Adding a column with parameter names, when possible, could help reading the table.*

A column has been added to group parameter names by the process they correspond to. It was complicated to include parameter names, as not all parameters are easily interpretable, but this may still help readability.

*Fig. S3 - x axis legend seems wrong. Do you mean "snow year" instead of "water year"? Please use the complete version of the caption presented in https://doi.org/10.5194/egusphere-2024-201-AC2.*
Thank you for catching this detail. We group consecutive December, January, and February months for one DJF season (labelled by the year that December belongs to). This is a "snow year" grouping convention.

*Fig. S5 - Please use the complete version of the caption presented in [https://doi.org/10.5194/egusphere-2024-201-AC2](https://doi.org/10.5194/egusphere-2024-201-AC2).*
The complete caption has been used.